# Respiratory Syncytial Virus-Associated Deaths among Children under Five before and during the COVID-19 Pandemic in Bangladesh

**DOI:** 10.3390/v16010111

**Published:** 2024-01-12

**Authors:** Md Zakiul Hassan, Md. Ariful Islam, Saleh Haider, Tahmina Shirin, Fahmida Chowdhury

**Affiliations:** 1International Centre for Diarrhoeal Disease Bangladesh (icddr,b), Dhaka 1213, Bangladesh; arif@icddrb.org (M.A.I.); abu.haider@icddrb.org (S.H.); fahmida_chow@icddrb.org (F.C.); 2Institute of Epidemiology, Disease Control and Research (IEDCR), Dhaka 1212, Bangladesh; tahmina.shirin14@gmail.com

**Keywords:** respiratory syncytial virus, RSV, COVID-19 pandemic, children, SARI death, Bangladesh

## Abstract

Respiratory syncytial virus (RSV) is a leading cause of acute lower respiratory infections in young children worldwide. RSV-associated deaths in children are underreported in Bangladesh. We analyzed hospital-based surveillance data on severe acute respiratory infections (SARIs) in under-five children before (August 2009–February 2020) and during the COVID-19 pandemic (March 2020–March 2022). Using the World Health Organization definition, we identified SARI cases in 14 tertiary-level hospitals. Nasopharyngeal and oropharyngeal swabs were collected for real-time reverse-transcriptase–polymerase chain reaction (rRT-PCR) testing of six respiratory viruses, including RSV. SARI deaths during the pandemic (2.6%, 66) were higher than pre-pandemic (1.8%, 159; *p* < 0.001). Nearly half of pandemic deaths (47%) had underlying respiratory viruses, similar to the pre-pandemic rate (45%). RSV detection in deaths was consistent pre-pandemic (13%, 20/159) and during the pandemic (12%, 8/66). Children aged < 6 months constituted 57% (16) of RSV-related deaths. Evaluating interventions like maternal vaccination and infant monoclonal antibody prophylaxis is crucial to address RSV, a major contributor to under-five SARI deaths.

## 1. Introduction

Respiratory syncytial virus (RSV) is a major cause of respiratory tract infections worldwide in infants and young children [1]. It is highly contagious and transmitted through respiratory droplets and touch. RSV-infected individuals often exhibit symptoms 4–6 days after infection. The symptoms generally include sneezing, coughing, wheezing, runny nose, increase in temperature, and decrease in appetite. Typically, these symptoms do not appear together; rather, they appear gradually. Infected newborns and infants may only symptomize diminished activity or lethargy, irritability, and breathing problems. Nearly every child faces RSV exposure before their second birthday [2]. Although most of the infected children are cured without any complications, some develop severe infections. The most common complications include bronchiolitis or inflammation of the small airways of the lungs, pneumonia, and eye and ear infections [2]

RSV poses a significant global health burden [3], predominantly in low- and middle-income countries (LMICs) [4]. Globally, 1 in every 50 deaths among under-five children is attributable to RSV, and in 2019 alone, there were around 3.6 million hospital admissions and 26,300 in-hospital deaths globally in children under five years due to RSV [4]. LMICs bear the brunt of the RSV burden, accounting for over 95% of RSV-acute lower respiratory infections and over 97% of related mortality in all age groups globally [4].

Bangladesh, an LMIC, faces a significant RSV burden, with approximately 90% of excess mortality during the RSV season attributed to RSV infections [5]. Hospitalization rates due to RSV are considerable, with RSV having the highest burden among both hospitalized (34%) and non-hospitalized (39%) cases in children under the age of five [6]. The economic impact of RSV is also substantial, with families spending a median direct cost of USD 62 and an indirect cost of USD 19 for hospitalization in 2010 [7]. The estimated median direct cost of RSV-associated hospitalization in children under five years was USD 10 million, with an indirect cost of USD 3.0 million in 2010, placing significant financial strain on affected families and the healthcare system [7].

The COVID-19 pandemic brought about significant changes in the epidemiology of RSV worldwide. Studies in developed countries reported delayed RSV peaks during the pandemic following an absence during its typical season [8]. Reduced transmission of RSV was observed due to non-pharmaceutical interventions, like lockdowns and school closings [9]. Furthermore, changes in the age distribution of RSV infections have been noted, with preschool-aged children being more affected than school-going children and newborns [10]. However, the specific impact of the pandemic on RSV epidemiology in LMICs, including Bangladesh, remains an important data gap that needs to be addressed.

Understanding the burden of RSV-associated under-five child deaths in Bangladesh, both during and before the COVID-19 pandemic, is critical for designing context-specific interventions and public health policies. Data on RSV-associated morbidity and mortality may guide effective strategies to reduce RSV burden. This study aimed to address these gaps and generate critical evidence on RSV-associated deaths among under-five children with SARI in Bangladesh, ultimately contributing to improved public health responses and better outcomes for the vulnerable under-five population.

## 2. Materials and Methods

### 2.1. Hospital-Based Surveillance Platform

We analyzed the data from the hospital-based influenza surveillance (HBIS) system to characterize SARI deaths. As a part of the National Influenza Centres (NIC), the HBIS was initiated in Bangladesh in 2007 in 12 tertiary care hospitals. The surveillance was conducted in a maximum of 14 hospitals at different geographical locations across Bangladesh over different time points (Figure 1); the number of sites ranged from 7 to 14 depending on various years. Since 2018, it has been operational in nine tertiary-care-level hospitals (seven public and two private) geographically distributed all over Bangladesh. The activities of the surveillance system are carried out jointly by the International Centre for Diarrhoeal Disease Research, Bangladesh (icddr,b) and the Institute of Epidemiology, Disease Control and Research (IEDCR) of the Government of Bangladesh (GoB), with technical support from the United States Centers for Disease Control and Prevention (US CDC). The inpatient capacity of the surveillance hospitals ranges from 500 to 1500 beds, with a 100–150% bed occupancy rate.

### 2.2. Study Population

For this study, we analyzed the data of participants enrolled from August 2009 to March 2022 in HBIS. Despite the ongoing pandemic in 2020 and subsequent pandemic control efforts, the surveillance remained active and continued its operations six days a week (Saturday to Thursday) during working hours (8:30 a.m. to 5:00 p.m.) by collecting data from the in-patient departments of the study hospitals. During the national holidays and the weekends (Friday), data collection was paused. In our study, the pre-pandemic period spanned from August 2009 to February 2020, and the pandemic period was considered from March 2020 to March 2022. Our study population contains all in-patient children under 5 years old admitted to the surveillance facilities during these periods.

### 2.3. Case Identification

The surveillance physicians and support staff screened and identified the severe acute respiratory infection (SARI) patients who met the WHO case definition of SARI, defined as an acute respiratory infection with subjective or measured fever of ≥38 °C and a history of cough with onset within the last 10 days from in-patient departments of medicine and pediatrics wards, coronary care units (CCUs), and specialized COVID-19 isolation wards established during the COVID-19 pandemic.

### 2.4. Data Collection

After identification, written informed consent was obtained from the parents or caregivers of the under-five children with SARI. The surveillance physicians then enrolled and performed a physical examination of all the under-five children with SARI. This was followed by the collection of data using a standardized surveillance form on a handheld computer. Real-time data transfers to our central server allowed for the development of algorithms that reported on primary missing variables and/or values in variables related to data quality. The form included demographic, clinical, history of comorbid conditions (e.g., diabetes, hypertension, cancer, asthma, chronic obstructive pulmonary disease, heart diseases), and available diagnostic findings of the patients. At the time of discharge, the outcome status (referral to another facility, partial recovery, full recovery, and in-hospital death) of the participants was recorded. The data were checked for a second time by the data management team and matched with the staff to ensure the rectification of the inaccuracies and to establish a robust system.

### 2.5. Sample Collection and Transportation

Maintaining all aseptic precautions, nasopharyngeal (NP) and oropharyngeal (OP) swabs were collected from the SARI patients after written informed consent from their parents or caregivers to participate in our study. Swabs were put into individual cryovials containing VTM and kept in a cool box for up to 30 min with a temperature between 2 °C and 8 °C. Both the NP swab and OP swab samples were labeled, packaged, stored in a nitrogen dry shipper (−150 °C) on site, and then transported to the virology laboratory of icddr,b, Dhaka, every week.

### 2.6. Laboratory Analysis

We tested all the in-hospital death cases for common respiratory viruses: RSV, adenoviruses, influenza, human metapneumovirus (HMPV), human parainfluenza viruses (HPIV), and SARS-CoV-2 (from March 2020); these were checked using real-time reverse-transcriptase–polymerase chain reaction (rRT-PCR). At icddr,b virology laboratory, InviMag Virus DNA/RNA Mini Kit (Invitek, STRATEC Molecular, Berlin, Germany) was used on the Kingfisher Flex 96 (Thermo Fisher Scientific, Waltham, MA, USA), an automated nucleic acid extraction tool to extract the viral nucleic acid from pooled NP and OP swab samples [11]. US-CDC provided us with the primers and probes. The primers and probes of the rRT-PCR assay were designed to detect the conserved regions of matrix genes. These are detected from the GenBank alignment sequences. The primers and probes used for detection of the the viruses of interest via rRT-PCR assay are given in the table below (Table 1) [12,13]. Specimen total nucleic acid (TNA) extract (5 µL), forward primer (0.5 µM), reverse primer (0.25 µM), and probe (0.05 µM) were used to prepare the reaction mixture. It was then amplified using an iCycler iQ^TM^ Real-Time Detection System (Bio-Rad, Hercules, CA, USA). Three cycling conditions were used: 1 10 min cycle at 48 °C, 1 5 min cycle at 95 °C, and then 45 cycles of 15 s at 95 °C followed by 1 min at 55 °C [14].

### 2.7. Data Analysis

We calculated descriptive statistics for all variables. Continuous variables were summarized using median and interquartile range (IQR) based on the distribution of the variables. We provided frequencies and proportions for categorical variables. We also used Chi-square and Fisher’s exact tests to compare the contribution of RSV in SARI mortality among under-5 children before and during the pandemic, where we considered a *p*-value < 0.05 statistically significant. We conducted the statistical analyses using Stata version 15, College Station, TX 77845, USA.

## 3. Results

### 3.1. Demographics of Study Participants

We enrolled 8923 under-5-year-old children with SARI during the pre-COVID-19 pandemic phase (August 2009–February 2020). The median age was 6 months (IQR: 2.5–12), and 67% were male (5956). During the pandemic period (March 2020–March 2022), 2570 children < 5 years were enrolled. The median age was found to be 6 months (IQR: 3–14): 65% males (1680). Almost 90% of the patients were younger than two years throughout the whole study period (Table 2).

### 3.2. Clinical Features of Study Participants

During both of the periods, the most common respiratory symptoms the patients reported were breathing difficulty (87%) and chest indrawing (76%). Other common clinical symptoms included runny nose and inability to drink. The parents or the caregivers reported to the hospital within an average of 2 days (IQR: 1–3 days) of the onset of symptoms. Almost all patients were discharged within 4 days (IQR: 3–6 days) of admission unless they were referred to a different facility or died.

### 3.3. Clinical Care of Study Participants

Over 90% of the children received antibiotics. None received any antiviral drugs. The necessity of supplementary oxygen increased more during the pandemic than in the pre-pandemic period. A total of nine patients required ICU support, and only one of them was from the pandemic period (Table 2).

### 3.4. Contribution of RSV in SARI Mortality Pre- and during Pandemic

We found that compared to the pre-pandemic period, the proportion of SARI deaths at the time of the pandemic was higher [((1.8%, 159) vs. 2.6%, 66); *p* < 0.001]. During the pre-pandemic period, 45% (71/159) of the death cases revealed respiratory viruses, whereas it was 47% (31/66) during the pandemic (Figure 2). Of 159 pre-pandemic deaths, RSV was predominantly detected (13%, 20). The other findings included adenovirus (8%, 12), HPIV (9%, 14), HMPV (6%, 10), and influenza (4%, 6). Further, nine (6%) death cases detected viral co-detections, including three (2%) co-detections with RSV. During the pandemic period, RSV (12%, 8) as well as adenovirus (12%, 8) comprised the largest proportions of the 66 pandemic deaths. The other detected viruses were SARS-CoV-2 (6%, 4), HPIV (6%, 4), influenza (3%, 2), and HMPV (1%, 2). We also detected co-detection with RSV and adenovirus (3%, 2) and with HPIV and adenovirus (3%, 2) (Figure 3).

### 3.5. Characteristics of RSV-Associated SARI Deaths among under-Five Children before and during the COVID-19 Pandemic

Before the onset of the COVID-19 pandemic, there were 20 recorded deaths in children under the age of 5 years attributed to severe acute respiratory syndrome associated with RSV. The median age of these cases was 3.5 months (IQR: 2.3–6 months), and 65% of the cases were male (13/20). Of these cases, 85% (17/20) were children aged < 1 year and 65% (13/20) were aged < 6 months. Co-morbid conditions were present in 10% of the cases (2/20).

During the COVID-19 pandemic, there were eight deaths in under-5-year-old children associated with RSV-SARI. The median age for these cases was 7.5 months (IQR: 2.5–13.5 months), and 25% were male (2/8). Among these cases, 75% (6/8) were children under 1 year of age, 38% (3/8) were under 6 months of age, and 13% (1/8) had at least one co-morbid condition.

RSV was solely detected in 57% (16) of the <6 months old, in 25% (7) of the 6–12 months old, in 11% (3) 1–2 years old, and 7% (2) of the 3–5 year olds among all the death cases. All cases exhibited a history of breathing difficulty.

## 4. Discussion

Our study showed that in both the pre-pandemic and pandemic periods, RSV was the major contributor to deaths among young children with SARI. Notably, we observed that nearly half of all SARI-related deaths in under-5-year-old children were associated with various respiratory viral pathogens, with RSV consistently responsible for a substantial proportion of these cases, exhibiting minimal variation between the two periods (pre-pandemic: 13%; during pandemic: 12%). These findings underscore the urgency of implementing measures to prevent these vaccine-preventable deaths.

The burden and impact of RSV infections among under-five children have not been well studied in Bangladesh. Our study, as far as we are aware, represents the first of its kind in Bangladesh, concentrating exclusively on mortality among children under the age of five linked to RSV. One of the main causative agents behind Bronchiolitis is RSV. According to the WHO estimates, RSV is responsible for about 60% of pediatric acute respiratory infections worldwide. Moreover, during the height of the viral season each year, RSV causes about 80% of lower respiratory tract infections (LRTIs) in infants under the age of one [15]. Pneumonia, another severe LRTI complication of RSV, may occur with or without bacterial co-detection. Around 40% of those who were admitted into the pediatric intensive care unit (PICU) due to severe RSV bronchiolitis were co-infected with respiratory bacteria and had a higher risk for bacterial pneumonia [16]. A prior study conducted in Bangladesh revealed a surge in unclassified pneumonia-related deaths (64%) in children < 2 years during the peak seasons of bronchiolitis [17].

Estimating RSV-related mortality in Bangladesh has proven challenging, primarily due to the absence of comprehensive, long-term systematic surveillance [18]. Our ongoing surveillance platform, the HBIS, has been consistently monitoring data, both before and after the onset of the COVID-19 pandemic. This platform enables us to systematically analyze deaths attributed to RSV and all respiratory viruses in a comprehensive manner. The findings from our study underscore the necessity for continued surveillance and further research to investigate the underlying factors associated with RSV-related mortality and to identify effective strategies for preventing childhood RSV infections and deaths in Bangladesh.

This study reveals a slight decrease in the total number of RSV-associated deaths among children under five years old with SARI during the COVID-19 pandemic. Previously, RSV-related deaths accounted for 20 out of 159 (13%) of all deaths, but during the pandemic, this figure slightly decreased to 8 out of 66 (12%). It is worth noting that the total number of SARI patients in children under five also decreased during this period. This trend of reduced RSV-associated morbidity and mortality during the pandemic has been observed in several other countries, including China [19], Austria [20], France [21], Brazil [22], and globally [23,24,25].

Apart from the concept of viral interference, the non-pharmacological interventions (NPIs) enacted during the COVID-19 pandemic, such as the use of face masks, frequent handwashing, social distancing, and lockdowns, may have played a role in the lower circulation of RSV. Washing hands frequently, putting on masks, and practicing respiratory hygiene habits helped to stop the spread of RSV along with COVID-19. These practices limited the circulation of virus-containing respiratory droplets. Curbing large gatherings and physical distancing also limited viral transmission. In addition to helping to avoid COVID-19, this action reduced the community spread of RSV [26].

When compared to the pre-pandemic period, our research indicates a shift in the median age of RSV-related deaths among children under five, increasing from 3.5 months (IQR 2.3–6) to 7.5 months (IQR 2.5–13.5) during the pandemic. The proportion of deaths due to RSV among children aged under six months decreased from 65% before the pandemic to 38% during the pandemic. This suggests that a higher percentage of older children, primarily over 6 months old, were succumbing to RSV infections during the pandemic compared to the pre-pandemic period. A study in France also reported a similar increase in the median age of RSV admissions among children [27]. An Australian study found that the median age of RSV-associated hospitalization in children significantly rose from 12.5 months in 2019 to 18.4 months after the COVID-19 pandemic NPIs were relaxed [28]. Various studies worldwide have reported that the decline in or near absence of RSV cases was followed by a delayed seasonal resurgence, accompanied by an increase in the median age of infection and death. We attribute this phenomenon to a significant cohort of older children who remained immunologically vulnerable due to the NPIs during the COVID-19 pandemic, and later, when these measures were relaxed, they came into contact with the virus [19,29,30,31].

To combat emerging RSV epidemics, it is necessary to promote personal hygiene practices and social distancing for sick individuals. It is neither practical nor feasible to maintain year-round generalized social distancing and lockdowns. But season, context, population-specific (e.g., kindergarten, elementary schools, etc.) mask mandates, temporary restrictions on large gatherings, and distancing should be taken into consideration as proven methods to curb the burden of RSV as well as all respiratory viral infections during future potential outbreaks.

Among other preventive measures, infant monoclonal antibody prophylaxis and RSV vaccination for mothers are noteworthy. Immunizing the pregnant mother during the second or third trimester of the pregnancy will boost the serum-neutralizing antibody response, the serum of which will eventually transfer to the fetus from the mother via the placenta during the prenatal period and via breastmilk during the postnatal period [32,33]. These maternal antibodies effectively produce immunity against RSV in RSV-naïve newborns and infants. The monoclonal antibodies provide pre-RSV exposure or prophylactic and passive immunization to infants, especially preterm or full-term at-risk babies from severe lower respiratory tract infection at the early stage of their lives [34]. These two options should also be evaluated to combat these RSV-associated premature deaths.

We observed that the presence of co-morbid conditions was 10% (2/20) before the pandemic and 13% (1/8) during the pandemic. Notably, a history of breathing difficulty was a common feature among children under five upon hospitalization. Due to the limited number of deaths, conducting a risk factor analysis was not feasible. However, future work should consider such analysis using a larger cohort and develop risk detection or prediction models to predict unfavorable outcomes. In resource-constrained settings like ours, the creation and utilization of clinical prediction tools can facilitate early disease severity detection, aid in diagnosis and prognosis, and offer clinical decision support [35,36].

Our study has several limitations. Firstly, we only tested SARI death cases. To gain a better understanding of the seasonality of RSV and the impact of COVID-19 on the seasonality and circulation of RSV, we need continuous year-round geographically representative surveillance. Secondly, we only used data from our hospital-based SARI surveillance and likely missed RSV cases that are non-SARI or non-medically attended due to healthcare-seeking behavior, where only 34% of the population receives healthcare from trained medical personnel [37]. We were also unable to conduct a risk factor analysis due to the low number of deaths captured through the hospital-based SARI surveillance system. Community-based surveillance can provide a more accurate insight into the RSV circulation and burden in Bangladesh.

## 5. Conclusions

RSV imposes a significant burden among under-five children, particularly in LMICs. Despite being a leading cause of under-five mortality in Bangladesh, the RSV burden has not been well studied. Our study has shown that both before and during the pandemic periods, RSV was a significant factor, leading to mortality in under-five children with SARI in Bangladesh. Despite some changes in RSV circulation due to the COVID-19 pandemic, it is expected that RSV circulation will re-emerge and cause local outbreaks in the near future. An increase in the median age of RSV-infected children indicated naturally unimmunized older children became vulnerable with the relaxation of NPIs.

Hospital- and community-based systematic surveillance is important to monitor RSV circulation and characterize RSV seasonality. National health authorities should promote personal respiratory hygiene. Season-, context-, and population-specific mask mandates and temporary social distancing should be implemented to minimize the community spread of RSV during potential seasonal outbreaks. In addition to that, other evidence-based measures, such as monoclonal antibody prophylaxis for infants and RSV vaccination for mothers, should be evaluated. These preventive interventions may help us combat RSV-associated unexpected premature deaths in the future.

## Figures and Tables

**Figure 1 viruses-16-00111-f001:**
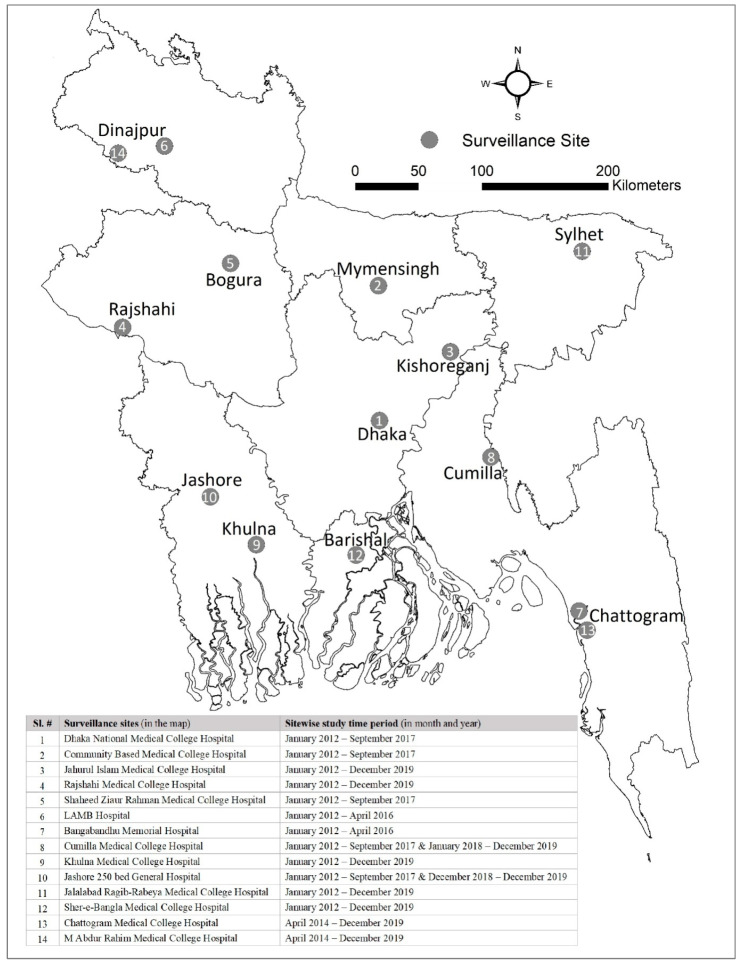
Location of the 14 surveillance hospitals of HBIS in Bangladesh.

**Figure 2 viruses-16-00111-f002:**
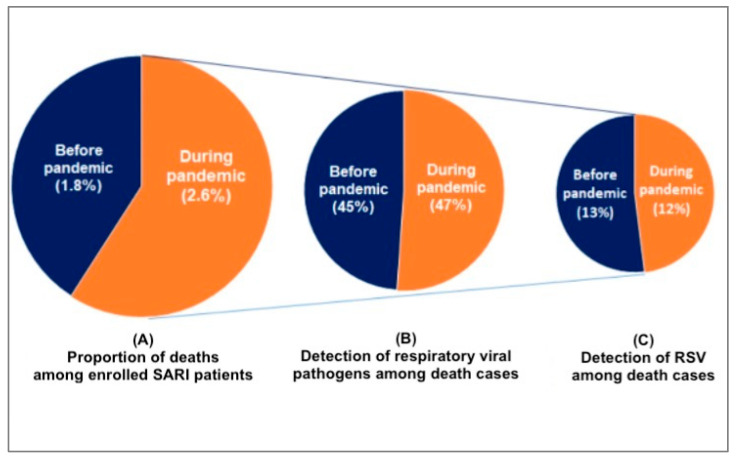
Proportion of deaths among the SARI patients aged < 5 years before and during the pandemic.

**Figure 3 viruses-16-00111-f003:**
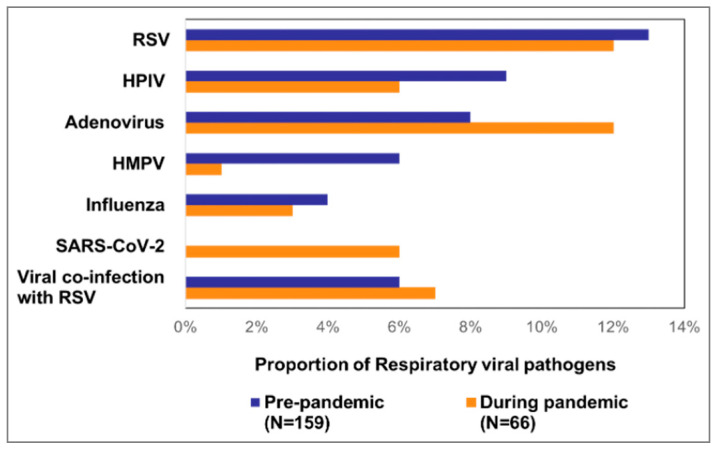
Respiratory viral pathogens detected among SARI death cases (aged ≤ 5 years) during the pre-pandemic and pandemic periods.

**Table 1 viruses-16-00111-t001:** Primers and probes used in this study for real-time reverse-transcriptase–polymerase chain reaction (rRT-PCR) assays.

Assay	Primer/Probe Sequence (5′–3′)
RSV	F, GGC AAA TAT GGA AAC ATA CGT GAA R, TCT TTT TCT AGG ACA TTG TAY TGA ACA G P, CTG TGT ATG TGG AGC CTT CGT GAA GCT
Influenza A	F, GAC CRA TCC TGT CAC CTC TGA C R, AGG GCA TTY TGG ACA AAK CGT CTA P, TGC AGT CCT CGC TCA CTG GGC ACG
Influenza B	F, TCC TCA ACT CAC TCT TCG AGC G R, CGG TGC TCT TGA CCA AAT TGG P, CCA ATT CGA GCA GCT GAA ACT GCG GTG
SARS-CoV-2	F, GACCCCAAAATCAGCGAAAT R, TCTGGTTACTGCCAGTTGAATCTG P, ACCCCGCATTACGTTTGGTGGACC
HMPV	F, CAA GTG TGA CAT TGC TGA YCT RAA 2 R, ACT GCC GCA CAA CAT TTA GRA A P, TGG CYG TYA GCT TCA GTC AAT TCA ACA GA
Adenovirus	F, GCC CCA GTG GTC TTA CAT GCA CAT C 2 R, GCC ACG GTG GGG TTT CTA AAC TT P, TGC ACC AGA CCC GGG CTC AGG TAC TCC GA
HPIV	F, AGT TGT CAA TGT CTT AAT TCG TAT CAA T 2 R, TCG GCA CCT AAG TAA TTT TGA GTT P, ATA GGC CAA AGA “T”TG TTG TCG AGA CTA TTC CAA

F, forward primer; R, reverse primer; P, probe; (P) = dP-CE (pyrimidine derivative); (A) = LNA-dA, (T) = LNA-dT (Locked Nucleic Acid (LNA) primers).

**Table 2 viruses-16-00111-t002:** Demographic, clinical and epidemiological characteristics of under-five children with severe acute respiratory infections (SARIs) before the COVID-19 pandemic (August 2009–February 2020) and during the COVID-19 pandemic (March 2020–March 2022) in Bangladesh.

Characteristics	SARI Patients Enrolled			
	Total SARI Patients	Before Pandemic	During Pandemic	*p*-Value
	N = 11,493 *n* (%)	N = 8923 *n* (%)	N = 2570 *n* (%)	
Demographic characteristics				
Age				
<2 Year	10,238 (89)	7980 (89.4)	2258 (88)	0.059
2–5 Years	1255 (11)	943 (10.6)	312 (12)	<0.001
Median age (IQR), years	0.5 (0.2–1)	0.5 (0.2–1)	0.6 (0.2–1.2)	<0.001
Sex (Male)	7637 (66.4)	5957 (66.8)	1680 (65.4)	0.188
Clinical Characteristics				
Runny nose	6925 (60.3)	5076 (57)	1849 (72)	<0.001
Difficulty of breathing	9959 (86.7)	7769 (87)	2190 (85)	0.014
Sore throat	51 (0.4)	51 (0.6)	0 (0)	-
Chest indrawing	8760 (76.2)	7126 (80)	1634 (63.6)	<0.001
Unable to drink	2835 (24.7)	2080 (23.3)	755 (29.4)	<0.001
Vomiting	1571 (13.7)	1446 (16.2)	125 (5)	<0.001
Lethargy	668 (5.8)	628 (7)	40 (1.6)	<0.001
Diarrhea	227 (2)	181 (2)	46 (1.8)	0.443
Duration of symptoms prior to admission in days; Median (IQR)	2 (1–3)	2 (1–3)	2 (1–3)	0.036
Length of hospital stay in days; Median (IQR)	4 (3–6)	4 (3–6)	4 (2–6)	<0.001
Co-morbid condition				
≥1 co-morbid condition (Self-reported)	11,406 (99.2)	8855 (99.2)	2551 (99.3)	0.907
Treatment received				
Antibiotic	7804 (91)	5398 (90)	2406 (93.6)	<0.001
Oseltamivir	0 (0)	0 (0)	0 (0)	-
Oxygen	3458 (41)	2232 (37.7)	1226 (47.7)	<0.001
Mechanical ventilation	4 (0.05)	4 (0.07)	0 (0)	-
ICU support (after admission in general ward)	9 (0.1)	8 (0.1)	1 (0.4)	-
Laboratory Results				
RSV ^†^	658 (29)	638 (29)	20 (17)	0.242
Influenza virus	898 (7.8)	698 (7.8)	200 (7.8)	1
SARS-CoV-2 ^ƛ^	6 (0.23)	0 (0)	6 (0.23)	-
HMPV ^†^	168 (7.3)	164 (7.5)	4 (3.4)	0.757
Adenovirus ^†^	155 (6.7)	141 (6.4)	14 (12)	0.43
HPIV ^†^	159 (6.9)	147 (6.7)	11 (9.4)	0.733
Co-detection with ≥2 respiratory viruses	150 (1.3)	136 (1.5)	14 (0.5)	0.762
Clinical outcome; Death	225 (2)	159 (1.8)	66 (2.6)	<0.001

^†^ A total of 2307 samples were tested for RSV, HMPV, Adenovirus and HPIV. ^ƛ^ A total of 2563 samples collected during March 2020–March 2022 were tested for SARS-CoV-2.

## Data Availability

The data presented in this study are available on request from the corresponding author. The data are not publicly available to ensure the protection of privacy.

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
