# Peer review of "Respiratory Syncytial Virus-Associated Deaths among Children under Five before and during the COVID-19 Pandemic in Bangladesh"

_viruses, 2024, doi:10.3390/v16010111_

Round 1

Reviewer 1 Report

Comments and Suggestions for Authors

What is the main question addressed by the research? Do you consider the topic original or relevant in the field? Does it address a specific gap in the field?

The paper shows the impact of RSV infection sequelae in low- and middle-income countries.  

What does it add to the subject area compared with other published material?  

There aren't many jobs on RSV infection sequelae in low- and middle-income countries in the literature.  

What specific improvements should the authors consider regarding the methodology? What further controls should be considered?

Methods need to be scaled up. The authors should add specifications regarding the PCR assays used: primer sequence, concentrations, master mix and reaction conditions (thermal profile etc.).

Are the conclusions consistent with the evidence and arguments presented and do they address the main question posed?  

Yes  

Are the references appropriate?  

Yes  

Additional comments on the tables and figures:  

The figures and tables are consistent with the data expressed in the result.

Reviewer 2 Report

Comments and Suggestions for Authors

The authors studied RSV-associated deaths under 5 years infants before/during COVID-19 pandemic in Bangladesh. Overall, the draft manuscript was well designed and described. The findings based on the work may contribute to understand infantile epidemiology of RSV infection and COVID-19.

Some minor issues were found. Please address them.

1. Line 38-39 of the sentence; Globally.... Please add a suitable reference.

2. Line 81; should amend as follow: 100-150%.

3. co-infections may not be suitable. please amend to co-detection(s).

Comments on the Quality of English Language

English quality is no problem.

Round 2

Reviewer 1 Report

Comments and Suggestions for Authors

I asked to include the PCR specifications in the methods. Only the method used for RSV has been inserted.
